# Anti-Inflammatory Function Analysis of *Lacticaseibacillus rhamnosus* CP-1 Strain Based on Whole-Genome Sequencing

**DOI:** 10.3390/biotech14020047

**Published:** 2025-06-07

**Authors:** Hanyu Chu, Lijie Zhou, Yanzhen Mao, Ren Liu, Jiaojiao Han, Xiurong Su, Jun Zhou

**Affiliations:** State Key Laboratory for Managing Biotic and Chemical Threats to the Quality and Safety of Agro-Products, School of Marine Science, Ningbo University, Ningbo 315832, China; 13462005107@163.com (H.C.); zhoulijie1113@163.com (L.Z.); mao2010111@163.com (Y.M.); 15102721800@163.com (R.L.); hanjiaojiao@nbu.edu.cn (J.H.); suxiurong@nbu.edu.cn (X.S.)

**Keywords:** *Lacticaseibacillus rhamnosus*, anti-inflammatory, bioinformatic analysis, gene function

## Abstract

*Lacticaseibacillus rhamnosus* (*L. rhamnosus*) is a safe probiotic with no side effects, providing benefits such as gut microbiota regulation and immune enhancement, making it highly valuable with strong potential. However, strains from different sources have unique traits, and whole-genome sequencing (WGS) helps analyse these differences. In this study, we used WGS to examine *L. rhamnosus* strains from mice with fish oil-treated smoking-induced pneumonia to better understand their biological functions and explore possible anti-inflammatory mechanisms. Methods: We isolated a strain, *Lacticaseibacillus rhamnosus* CP-1 (*L. rhamnosus* CP-1), from mice intestines where fish oil alleviated smoking-induced pneumonia. Identification of probiotic-related genes by WGS and characterised the strain’s probiotic properties. Results: *L. rhamnosus* CP-1 has a single circular chromosome (2,989,570 bp, 46.76% GC content) and no plasmids. COG, GO, and KEGG databases revealed genes linked to carbohydrate metabolism. The CAZy database identified GH25 lysozyme and PL8 polysaccharide lyase genes. KEGG highlighted an antimicrobial peptide ABC transporter permease, while TCDB noted the ABC-type antimicrobial peptide transporter (the main active transport component). KEGG also showed 10 genes for terpenoid skeleton biosynthesis and 5 for keto-glycan unit biosynthesis. Additionally, *L. rhamnosus* CP-1 carries metabolic regulators and bacteriocin-related genes. Conclusions: Whole-genome sequencing analysis revealed that *L. rhamnosus* CP-1 has carbohydrate utilisation and potential anti-inflammatory effects at the molecular level. Potential functional genes include carbohydrate transport and hydrolase, antimicrobial peptide ABC transporter and its osmotic enzyme components, bacteriocin immune protein, terpenoid skeleton, and keto-glycan synthesis.

## 1. Introduction

In our previous study, the anti-inflammatory and antioxidant functions of tuna oil (TO) in cigarette smoke (CS) exposure-induced lung inflammation in mice were demonstrated. At the genus level, the abundance of *Lacticaseibacillus* in the CS-induced model group was reduced compared with the blank control group. The abundance of *Lacticaseibacillus* increased in the experimental group with different doses of fish oil when compared with the CS-induced pneumonic mice. At the species level, the abundance of *Lacticaseibacillus rhamnosus* (*L. rhamnosus*) increased after treatment with different doses of TO [1]. The anti-inflammatory properties of *Lacticaseibacillus* have been shown to enhance the biosynthesis of serum IL-10, while concurrently inhibiting the production of TNF-α, IL-6, and interleukin-12 (IL-12). This dual action contributes to the subsequent mitigation of inflammatory responses in animals [2,3]. *L. rhamnosus* is a widely studied *Lacticaseibacillus* in both domestic and international research. Additionally, it is a facultative anaerobic, non-spore-forming, Gram-positive bacteria with long or short rods. It is classified within the genus *Lacticaseibacillus* under the *rhamnose* subspecies. It is acid-resistant, resistant to pancreatic juice and bile salts, and resistant to a variety of antibiotics and other biological characteristics [4]. *L. rhamnosus* is a symbiotic microorganism in the intestinal system of humans and animals. Its advantage lies in its high intestinal adhesion rate and strong colonisation. Additionally, it is beneficial for improving the host’s systemic immune response, and is often used to enhance the body’s immunity and disease prevention and treatment [5]. *L. rhamnosus* earned the prestigious designation of being generally recognised as safe (GRAS) by the United States Food and Drug Administration. It has also been included in the European Food Safety Authority’s Qualified presumption of safety (QPS) registry [6,7]. A growing body of evidence demonstrates that *L. rhamnosus* and its metabolites can modulate immune cells, such as M1 macrophages and T lymphocytes, effectively suppressing the production of pro-inflammatory cytokines including tumour necrosis factor-alpha (TNF-α), interleukin-1β (IL-1β), interleukin-6 (IL-6), and interleukin-2 (IL-2). Concurrently, these microbial components enhance the generation of anti-inflammatory cytokines such as interleukin-10 (IL-10), thereby regulating inflammatory responses and maintaining immune homeostasis through this dual mechanism of cytokine balance modulation [8,9,10]. However, recent research has shown that probiotics’ function is highly strain-specific. Their biological effect cannot be estimated at the species level and necessitates a strain-by-strain assessment [11]. This is one of the reasons why researchers continue to screen new probiotic strains with potential health benefits.

Comparative genomic analysis stands as a staple in bioinformatics, serving as a powerful tool to unravel the intricate connections between bacterial strains and their respective origins. It also facilitates the assessment of gene distribution within specific species, thereby shedding light on the phenotypic traits exhibited by the organisms under study [12,13]. Simultaneously, compared with other types of molecular biology technology, the research results of genomics have wider coverage and deeper analysis. With the development of technology in recent years, whole-genome sequencing (WGS) technology based on all genetic materials emerged. This method can fully mine the gene information of microorganisms, annotate their functional characteristics, and predict the relevant metabolic pathways, laying the foundation for studying the classification relationship and genetic progress of microorganisms [14]. In the early 21st century, Bolotin et al. first sequenced the whole genome of the first *Lacticaseibacillus lactic* IL1403 (*L. lactic* IL1403), thereby establishing the comprehensive availability of in-depth genetic and physical maps for the *L. lactic* IL1403 genome [15]. Following this seminal work, an array of *Lacticaseibacillus* species have undergone extensive sequencing and in-depth exploration, expanding our understanding of these microorganisms [16]. All the genetic material information of strains and for classifying the coding genes were obtained through WGS, which plays an essential part in the comprehensive analysis of *Lacticaseibacillus* bacteria. Therefore, in this study, the probiotic-related genes and probiotic characteristics of *L. rhamnosus* strains were isolated from the guts of mice with smoking-induced pneumonia relieved by fish oil and were deeply understood through WGS to better understand their potential biological functions and information and explore their possible anti-inflammatory mechanism.

## 2. Materials and Methods

### 2.1. Bacterial Strains Selection and Cultivation Conditions

The *L. rhamnosus* strain was successfully isolated from the gastrointestinal tract of mice that had been treated with fish oil to alleviate smoking-induced pneumonia and stored in the microbiology laboratory of Ningbo University (Ningbo, China). It was scribed and inoculated in de Man, Rogosa, and Sharpe agar solid medium (MRS; Qingdao Hope Bio-Technology Co., Ltd., Qingdao, China), as well as incubated at 37 °C for about 12 h under constant temperature and anaerobic conditions. Following this incubation, a single colony was selected from the medium and transferred to a 10-millilitre volume of MRS liquid medium. This liquid culture was subsequently incubated at 37 °C under anaerobic conditions until OD600 = 0.4–0.6.

### 2.2. Genomic DNA Extraction

The bacterial suspension was subjected to centrifugation at 6000 rpm for a duration of 5 min to obtain bacterial precipitation. Following this, the cells were meticulously washed and re-precipitated twice using 0.10 mol/L phosphate buffered saline (PBS) buffer, adjusted to a pH of 7.4, before undergoing further centrifugation. The precipitate was resuspended in 0.10 mol/L PBS, and the genomic DNA of the strain was extracted using a DNA extraction kit (Omega Bio-tek, Norcross, GA, USA). The purity and concentration of the extracted DNA were meticulously evaluated both quantitatively and qualitatively using the Qubit 4.0 Fluorometer (Q33226, Invitrogen, CA, USA) and the Nanodrop 2000 UV–vis spectrophotometer (Thermo Scientific, Wilmington, NC, USA). The PCR-amplified DNA fragments were then subjected to purification, following which they were dispatched to Shanghai Personalbio Technology Co., Ltd. (Shanghai, China) for comprehensive sequencing analysis.

### 2.3. Whole-Genome Sequencing

#### 2.3.1. Genomic Sequencing and Assembly

The WGS strategy was employed to construct libraries featuring various inserted DNA fragments. These libraries were subsequently sequenced utilising both next-generation sequencing (NGS) technology on the Illumina NovaSeq sequencing platform (Personalbio, Shanghai, China) and the cutting edge third-generation single-molecule sequencing technology facilitated by the Oxford Nanopore ONT platform (Personalbio, Shanghai, China). The third-generation single-molecule sequencing results were spliced to obtain the counting sequence. Subsequently, the second-generation sequencing data were utilised to refine and splice the third-generation results, yielding a more accurate and complete genome sequence.

#### 2.3.2. Analysis of Functional Components

The gene prediction for the WGS was conducted using the GeneMarkS (version 1.14) software [17]. The identification of tRNA genes within the genome was achieved with the tRNA scan-SE (version 2.0.9) tool [18]. The Barrnap (version 0.9) software was utilised to predict the presence of rRNA genes, while the identification of the remaining non-coding RNAs was predominantly achieved through a comparative analysis with the Rfam database (version 14.0) [19]. The RepeatMasker (version 4.0.5) software was used to identify repetitive sequences under homology annotation. The CRISPR finder (version 1.2) tool was employed to identify clustered regularly interspaced short palindromic repeats (CRISPRs) throughout the entire genome sequence [20].

#### 2.3.3. Genomic Subsystems Analysis

The presence of prophages in the genome was predicted using PhiSpy (version 4.2.21) [21]. Gene islands present in the genome were predicted by IslandViewer 4 (version 4.0) [22]. BLAST (version 2.14.0) software was used to predict the presence of virulence factor-related genes and antibiotic resistance-related genes in the genome [23,24,25]. The prediction of carbohydrate-active enzymes (CAZy) in the genome was achieved with the aid of Hmmscan (version 3.3.2) software, which specialises in identifying sequence motifs associated with these enzymes [26,27]. Additionally, the identification of secreted proteins was conducted using the SignalP (version 5.0) and TMHMM (version 2.0c) tools, which are recognised for their effectiveness in predicting protein secretion pathways and transmembrane helices, respectively [28,29].

#### 2.3.4. Functional Genome Annotation

Following rigorous quality control of the sequencing data and the meticulous elimination of low-quality sequences, the functional annotation of protein-coding genes is conducted. The predicted protein-coding genes are compared with proteins contained in various databases, and the results are filtered to predict and analyse the gene function of the strain at the molecular level. The databases for predicting gene function include the Kyoto encyclopedia of genes and genomes (KEGG), clusters of orthologous groups (COG), gene ontology (GO), the Transport classification database (TCDB), and the Swiss-Prot protein sequence database.

### 2.4. Comparative Genomic Analysis

Based on the chromosomal sequences derived from the genome assembly, we employed the fastANI (version 1.34) software to identify 20 phylogenetically proximate species within the database, subsequently performing phylogenetic reconstruction of sample core genes through the UBCG (version 3.0) software. Functional genes were retrieved from the KEGG database. Then, *L. rhamnosus* CP-1 was compared and analysed with 20 other strains. The Pearson correlation coefficient was used to calculate the genetic correlations among different strains. A clustering algorithm was adopted to cluster the strains, and GraphPad Prism (version 9.5.1) was employed to draw the graph.

### 2.5. Statistical Analysis

Data were analysed using SPSS Statistics 22.0 software (IBM Corp., Armonk, NY, USA). Graphs were plotted using Origin (v8.5) software (OriginLab Corporation, Northampton, MA, USA).

### 2.6. Data Availability Statement

The data sets utilised in the present study are archived in online repositories. The specific repository and its associated accession number(s) are detailed under PRJNA1069864.

## 3. Results

### 3.1. Culture and Genomic DNA Extraction of L. rhamnosus CP-1

The isolated strain was characterised through 16S rRNA gene sequencing, which confirmed a remarkable >99% sequence similarity with *L. rhamnosus*, as determined by BLAST (version 2.16.0) analysis. Consequently, the strain has been designated as *Lacticaseibacillus rhamnosus* CP-1 (*L. rhamnosus* CP-1).

Upon cultivation on MRS medium, the colonies of *L. rhamnosus* CP-1 presented distinct characteristics, exhibiting white, round, and convex morphology with well-defined, neat edges, and featuring smooth and moist surfaces (Figure 1a). After DNA extraction from the *L. rhamnosus* CP-1 isolate, a quality assessment was performed via 1.20% agarose gel electrophoresis (Figure 1b). The analysis revealed that the extracted DNA was of high integrity, exhibiting no significant degradation or contamination, thereby satisfying the stringent criteria necessary for library construction.

### 3.2. Genome Sequencing and Assembly

The quality control assessments of the sequencing data revealed that the overall quality of the data was exceptionally high (Figure A1, Table A1). Assembling and integrating the third-generation and second-generation sequencing data, and the results show that it is a cyclic sequence with a GC of 46.76%, which is of a qualified quality and can be used for subsequent components and prediction based on the assembling results (Table A2).

### 3.3. Genomic Component Prediction

#### 3.3.1. Genome Overview

The analysis revealed that the genome of *L. rhamnosus* CP-1 consisted of a completely dressed chromosome. The genome of the strain encompasses a total length of 2,989,570 bp, with 2767 coding genes that collectively occupy 2,529,300 bp, representing 84.60% of the genome’s composition. Importantly, the genome is devoid of any plasmids (Table A3). The graphical map of this genome was drawn using review software, including all genes, COG annotated genes, prophage, CRISPR, non-coding RNAs, GC content, and GC skew (Figure 2).

#### 3.3.2. Non-Coding Gene Prediction

The genomic analysis of *L. rhamnosus* CP-1 reveals a relatively straightforward profile of non-coding RNAs (Table 1). Among them, there are 59 tRNAs, with an average length of 75 bp and a total length of 4467 bp, accounting for 0.15% of the genome sequence. There are three types of rRNA: 5s rRNA, 16s rRNA, and 23s rRNA. There were five in each type, and the average length was 112 bp, 1566 bp, and 2914 bp, respectively. The total length was 560 bp, 7830 bp, and 14,570 bp, respectively. It accounted for 0.02%, 0.26%, and 0.49% of the genome sequence. There are 39 ncRNAs, with an average length of 176 bp and a total length of 6878 bp, accounting for 0.23% of the genome sequence.

#### 3.3.3. Repeat Sequence Prediction

A high prevalence of long terminal repeat sequences was observed, with a total of 76 instances annotated within the genome (Figure 3). The numbers of long dispersed repeat sequences, DNA transposons, short dispersed repeat sequences, and satellite RNAs were 26, 23, 10, and 7, respectively. Additionally, eight sequences remain unclassified. Notably, the genome does not contain any small RNAs, simple repeats, or sequences of low complexity.

#### 3.3.4. Prophage Prediction

The predicted results show that the number of prophages is nine, with a total length of 93,260 bp, accounting for 0.02% of the genome sequence, and the average length is 31,086.70 bp (Table 2).

#### 3.3.5. Gene Island Prediction

A genome island is a gene cluster that can be transferred horizontally in a microbial genome. According to different functional genes, it can be divided into a resistance island, metabolic island, symbiotic island, virulence island, and so on. The results show that 19 gene islands were predicted in the genome of the *L. rhamnosus* CP-1, and the size of the gene islands ranged from 5010 to 52,569 bp (Figure 4).

Among them, the genome island with a starting position of 455,612–460,338 bp was annotated with a total of three genes, a PRD domain-containing protein (WP_005691528.1), β-glucoside-specific phosphotransferase (PTS) system transporter subunit IIABC (WP_005691530.1), and glycoside hydrolase family 1 protein (WP_005691532.1). The genome islands located at 2,517,215–2,562,618 bp were annotated metabolic regulatory proteins and bacteriocin-related genes, and they included GHKL domain-containing proteins (WP_019728291.1), bacteriocin immunity proteins (WP_005692260.1, WP_005692238.1), class IIb bacteriocin, lactation A/cerein 7B family (WP_005692236.1), and bacteriocin (WP_005686851.1). Furthermore, genes for the peroxide stress regulator PerR (AGP73498.1) and the ThiJ/PfpI family protein (AGP69873.1) were also annotated within this genome island.

### 3.4. Genome Subsystem Analysis

The genome of *L. rhamnosus* CP-1 contained 11 virulence factors, which constitute just 0.40% of its total gene complement (Table 3). These virulence factors encompass genes for adhesion, such as Lap (VFG006717), EF-Tu (VFG046465), GroEL (VFG012095), and EfaA (VFG002165), as well as three genes linked to stress response—clpC (VFG000077), clpE (VFG000080), and clpP (VFG000079). Furthermore, the genome features four genes related to immune regulation, encompassing those for a hyaluronic acid capsule (VFG000964), two genes for a capsule (VFG002190/VFG048830), and one gene for lipopolysaccharide (LPS) (VFG047039).

There are 20, 11, 2, and 27 genes annotated in the antibiotic resistance, the antibiotic target, the antibiotic biosynthesis, and the total resistance genes, and they accounted for 0.72%, 0.40%, 0.07%, and 0.98% of the total genes, respectively (Figure 5a). Most of the antibiotic resistance genes were related to daptomycin, aminocoumarins, and fluoroquinolones. No antibiotic resistance genes or virulence genes were found in prophages. One virulence gene associated with bacteriocin and four antibiotic resistance genes (three linked to bacteriocin and one related to ABC transporter) were identified within genomic islands.

In terms of enzymatic functions, 112 gene matches were identified in *L. rhamnosus* CP-1 that correspond to related enzymes. This includes 49 glycoside hydrolases (GH), 37 glycosyltransferases (GT), 17 carbohydrate esterases (CE) genes, 4 carbohydrate-binding modules (CBM) genes, 3 genes for auxiliary activities (AA), and 2 genes for polysaccharide lyases (PL). These categories represent 43.75%, 33.04%, 15.19%, 3.57%, 2.66%, and 1.79% of the total enzymes, respectively (Figure 5b). Within the GH group, genes associated with lysozyme (EC 3.2.1.17) were annotated, and genes associated with hyaluronate lyase (EC 4.2.2.1), chondroitin AC lyase (EC 4.2.2.5), xanthan lyase (EC 4.2.2.12), and chondroitin ABC lyase (EC 4.2.2.20) were annotated within the PL8 group.

### 3.5. Genome Functional Annotation

#### 3.5.1. KEGG Database Notes

A total of 2700 genes were annotated in the KEGG database, accounting for 97.58% of the WGS of *L. rhamnosus* CP-1. Within the KEGG BRITE database, a comprehensive annotation reveals that 444 genes are associated with genetic information processing, while 387 genes are linked to signalling and cellular processes. Additionally, 177 genes have been annotated to the metabolism category. In the KEGG pathway database, the metabolic pathways of carbohydrate metabolism and amino acid metabolism stood out, with significant annotations to 310 and 147 genes, respectively. Furthermore, within the environmental information processing pathway, there were 168 genes dedicated to membrane transport processes. Within the immune system category, TrxA genes associated with the NOD-like receptors (NLRs) signaling pathway are annotated. In the metabolism category, there are polyketide glycan units, terpenoid skeletons, and polyketide glycan unit biosynthesis (Figure 6a). Genes associated with the Clp protease family, such as clpB/C/L/X/P (K03695/K03696/K04086/K03544/K01358), as well as those related to glutamate synthetase including GlnA/P/Q (K01915/K10040/K10041), have been annotated. Additionally, genes for the oppA/B/C/D/F (K15580/K15581/K15582/K15583/K10823) family, which are connected to the oligopeptide transport system and osmotic protein, and the dltA/B/C/D (K03367/K03739/K14188/K03740) genes, which are related with cationic antimicrobial peptides (AMPs), were annotated to the KEGG database.

#### 3.5.2. GO Database Notes

The GO annotation for *L. rhamnosus* CP-1 mainly included three categories of biological process, molecular function, and cellular component, and 1905, 1841, and 588 genes were annotated, respectively. Among the annotations, the category of biological processes is predominant, encompassing a substantial number of genes across multiple subcategories. Notably, 729 genes are associated with cellular nitrogen compound metabolism, followed by 616 genes in biosynthesis, 517 genes in small molecule metabolism, and 200 genes in carbohydrate metabolism. In the molecular function category, there is a substantial representation, with 635 genes annotated for ion binding, 298 genes for DNA binding, 199 genes for oxidoreductase activity, and 147 genes for kinase activity. Regarding cellular components, a large proportion of genes are annotated for glycosylated structures, with 495 genes in intracellular locations, 389 genes in the cytoplasm, and 214 genes in the plasma membrane (Figure 6b).

#### 3.5.3. COG Database Notes

A total of 2334 genes in the *L. rhamnosus* CP-1 genome were successfully annotated with COG functions, which accounted for 84.35% of its WGS. These annotations reveal that the protein functions of the strain are predominantly distributed across three major categories: 10.12% are involved in carbohydrate transport and metabolism (G), with 280 genes, 7.05% are related to transcription processes (K), encompassing 195 genes, and 6.25% are engaged in amino acid transport and metabolism (E), consisting of 173 genes. Additionally, 131 genes are annotated under cell wall/membrane/envelope biogenesis (M), which constitutes 4.73% of the total. Furthermore, there are 21 genes annotated as histidine kinase genes, which are associated with the signal transduction mechanism (Figure 6c). A considerable number of genes associated with the dehydrogenase functions and the PTS system were also annotated within the COG database. Notably, the gene dltD, which is associated with the D-alanyl-lipoteichoic acid biosynthesis protein was also annotated.

#### 3.5.4. TCDB Database Annotation

Within the TCDB database, 568 genes were annotated, accounting for 20.53% of the WGS of *L. rhamnosus* CP-1. Primary active transporters were the most numerous, with a total number of 254 annotated, and 98 genes were annotated to group transporters, 97 for electrochemical potential-driven transporters, 56 for fully characterised transporter systems, 29 for channels/pores, 22 for cofactors involved in transport, and 12 for transmembrane electron carriers (Figure 6d). Among them, the ABC-type anti-microbial peptide transporter and ATP-binding protein gene (Bd3816) were annotated in the group transporters. The sugar-specific component transporter genes of glucose, sorbose, cellobiose, fructose, and lactose in the PTS system were annotated in the group transporters.

#### 3.5.5. Swiss-Prot Database Annotation

Redundant sequences were minimised in Swiss-Prot, and a total of 1927 unigenes were annotated, accounting for 69.64% of the *L. rhamnosus* CP-1 WGS (Table 4). The annotated genes contained UvrABC system protein A/B/C genes (uvrA/B/C), sugar utilisation genes such as PTS system sorbose-specific EIIA, EIIB, EIIC, and EID component genes (sorA, sorB, sorC, and sorD), lactose-specific EIICB, EIIA component genes (lacE, lacF), and so on. In addition, the annotation also encompasses genes involved in protein transport and metabolism, such as thiamine transporter genes (thiT), triggering factors (tig), bifunctional protein genes (folD, pyrR1), and so on.

### 3.6. Comparative Genomics of Lactobacillus rhamnosus CP-1 Strain

According to the results of phylogenetic tree analysis, the phylogenetic relationship between *L. rhamnosus* CP-1 and other strains showed obvious hierarchical characteristics, which provided important clues for analysing the evolutionary history of this strain. *Lacticaseibacillus rhamnosus* NCTC13764 and *L. rhamnosus* CP-1 clustered in the same branch, and the support value reached 92. The higher support value indicated that there was a close genetic relationship between them at the core gene level, and this close clustering relationship probably reflects that they have a closer common ancestor or experienced fewer genetic divergence events during evolution. Strains such as *Lacticaseibacillus rhamnosus* VSI43 and *Lacticaseibacillus rhamnosus* LMG 23550 were in a branch with higher support values with *L. rhamnosus* CP-1, showing a relatively close evolutionary relationship, and based on this, it can be inferred that these strains may belong to the same evolutionary branch or subspecies, and they may share a similar genetic background and evolutionary path during long-term evolution; therefore, there is a high consistency in the composition and arrangement of core genes. However, other strains, such as *Lacticaseibacillus rhamnosus* BIO6870 and *Lacticaseibacillus rhamnosus* 484, were distributed in distant nodes and had significantly lower support values, which suggested that they were genetically different from *L. rhamnosus* CP-1, and these differences may be due to different selection pressures during evolution, differences in horizontal gene transfer events, or long-term geographical isolation, which may lead to their different subpopulations (Figure 7a). The results of functional gene cluster analysis show that most strains showed a dark blue color with *L. rhamnosus* CP-1 at many gene loci, such as agaD, acmA, etc. A small number of white loci means that there is a weak correlation with the *L. rhamnosus* CP-1 strain at the corresponding gene locus, and there are differences in genetic characteristics. Strains such as *Lacticaseibacillus rhamnosus* BFE5264 and 4B15 were similar to *L. rhamnosus* CP-1 in most genetic loci, but showed white in some loci, showing some genetic differences (Figure 7b).

## 4. Discussion

The earliest isolated *Lacticaseibacillus rhamnosus* GG (LGG) has been recognised as a quintessential probiotic strain. Fundamentally, the probiotic properties of various strains are intimately tied to their genome, with their growth characteristics and nutritional functions being dictated and shaped by the intricate guidance of their gene sequences. *L. rhamnosus* is a non-toxic probiotic with no side effects and has biological functions such as regulating the gut microbiota, preventing and treating diarrhhoea, toxin elimination, and enhancing the body’s immunity, which is of high value and developmental prospects [30,31]. However, strains derived from various sources exhibit distinct characteristics, and WGS enables us to comprehend these differences more thoroughly. For probiotics used in disease relief or treatment, higher safety is a prerequisite. Sequencing analysis revealed that the *L. rhamnosus* CP-1 genome lacks genes encoding transmissible and infectious virulence factors. Given that these resistance genes do not possess the potential for horizontal transfer to other strains, they pose no significant safety concerns. In line with the classification by Ochman and Davalos, the genome size of *L. rhamnosus* CP-1 is considered medium-sized, which is typically associated with a robust metabolic capacity, high tolerance levels, and the ability to adapt to a variety of ecological niches [32,33]. *L. rhamnosus* CP-1 is devoid of plasmids, thereby eliminating the potential for the transfer of antibiotic resistance and virulence genes among bacteria via plasmid-mediated mechanisms. This characteristic makes it a safer option when compared to other strains that carry plasmids. Referring to the information available in public databases, *L. rhamnosus* CP-1 exhibits a distinct genetic profile compared to the *Lacticaseibacillus rhamnosus* 1.0320 (*L. rhamnosus* 1.0320) previously known. While *L. rhamnosus* 1.0320 boasts a genome length of 2.90 Mb and contains 2736 genes, *L. rhamnosus* CP-1 has a higher number of operons for fermentation OFRs and dedicated carbohydrate utilisation proteins. This genetic advantage positions *L. rhamnosus* CP-1 to more effectively engage in glycolysis and other probiotic functions [12]. In addition, according to the results of genome alignment, it was indicated that the unique evolutionary status of *L. rhamnosus* CP-1 was revealed, which formed a close association cluster with a specific closely related strain on the core gene evolutionary tree, indicating that *L. rhamnosus* CP-1 is an important genetic resource for *L. rhamnosus*, as it not only reflects the high genetic similarity with some strains, but also clarifies the genetic boundaries with other strains, and this study provided a solid theoretical basis for further study on the phylogeny, population differentiation, and evolution of functional genes of the strain. At the same time, functional gene cluster analysis showed that strain zero exhibited strong correlation and consistent genetic characteristics with *L. rhamnosus* CP-1 at many gene loci (such as agaD, ACMA, etc.). A small number of genetic loci showed weak correlation, indicating that there were differences in genetic characteristics. Most of these genes are related to the synthesis of sugar-specific components of glucose, sorbose, cellobiose, fructose, and lactose, which can produce SCFAs through related metabolic pathways, and they are also involved in the regulation of glucose metabolism and play a variety of anti-inflammatory and antibacterial-related physiological functions. This genetic difference may affect the functional characteristics of strains, which provides important clues for further study of genetic diversity and functional differences between strains.

Furthermore, ongoing research into the molecular underpinnings of the functional disparities within *L. rhamnosus* encompasses a range of gene-associated processes. These include genes are linked to fimbriae proteins, carbohydrate transport and metabolism, the biosynthesis of extracellular polysaccharides, bacteriocin production, restrictive modification systems, and bacterial defence mechanisms such as the CRISPR-Cas system. Regarding prebiotic functions, the majority of these are associated with carbohydrate metabolism, surface carbohydrate modifications, and surface proteins. *L. rhamnosus* is capable of fermenting a diverse array of carbohydrates, thereby harnessing metabolic energy from these substrates [34]. According to the CAZy database annotations, glycosidase genes were found to be the most abundant, and glucose, fucose, lactose, galactose, raffinose, and other sugars could be used. In contrast, coenzyme genes are annotated in the least number, yet they represent a broad class of redox-active enzymes that are indispensable for the intricate processes of carbohydrate metabolism [35]. The sugar-specific component transporter genes of glucose, sorbose, cellobiose, fructose, and lactose in the PTS system were annotated in the TCDB database. They are closely related to carbohydrate transport and catabolism. The diversity of carbohydrate metabolism genes enables probiotics to efficiently degrade dietary fibers (e.g., cellulose, pectin, and inulin) that are indigestible to the host, converting them into short-chain fatty acids (SCFAs, including acetate, propionate, and butyrate). These metabolites serve not only as crucial energy substrates for intestinal epithelial cells, but also exert multiple physiological functions: lowering intestinal pH to inhibit pathogenic bacterial proliferation, activating host immune signalling pathways (e.g., G protein-coupled receptors GPR41/43) to modulate local inflammatory responses and barrier functions, and systemically influencing host metabolism through circulatory transport [36]. Furthermore, the superior carbohydrate utilisation capacity grants probiotics ecological competitiveness in the gut niche, allowing them to outcompete pathogens through nutrient sequestration, antimicrobial secretion, and adhesion-mediated colonisation, thereby maintaining microbial homeostasis. Certain genes may additionally participate in metabolising mucin-derived oligosaccharides within the intestinal mucus layer, facilitating mucus renewal and enhancing barrier integrity, while pathway byproducts such as B vitamins supplement host nutritional requirements. The synergistic effects of these functions not only enhance probiotic adaptability in the complex gut environment, but also extend their health benefits through systemic metabolic regulation (e.g., SCFA-mediated modulation of hepatic and adipose tissue metabolism), potentially ameliorating metabolic disorders, including obesity and diabetes [16,34,35,36]. Notably, the enrichment of such genes may reflect co-evolutionary adaptations between probiotics and hosts during long-term symbiosis. However, comprehensive validation integrating transcriptomic/proteomic analyses and in vivo models remains essential to elucidate the functional activity and physiological contributions of these genetic elements. It is important to note that bacteriocins constitute a class of compounds that are synthesised within bacterial ribosomes and possess antibacterial properties. Typically, these bacteriocins do not exhibit antibacterial activity against the bacteria that produce them. The osmotic component of the antimicrobial peptide ABC transporter, various bacteriocin immunoproteins, and glycosidase lysozyme were annotated in the whole genome of *L. rhamnosus* CP-1. Meanwhile, in the CAZy database, four genes encoding GH25 lysozyme (EC 3.2.1.17) were annotated. Lysozyme, the seminal antibacterial peptide to be identified, stands out as a natural enzyme that possesses potent antimicrobial properties. It selectively targets and disrupts microorganisms, particularly Gram-positive bacteria, by selectively hydrolysing the 1,4-beta linkages that bind N-acetylmuramic acid to N-acetylglucosamine within the bacterial cell wall structure [37]. These findings suggest that *L. rhamnosus* CP-1 has possible potential antibacterial activity at the molecular level, but the specific antibacterial properties need to be further verified by in vitro related antibacterial experiments.

Notably, hyaluronan lyase plays a crucial role in degrading hyaluronic acid into smaller hyaluronan oligosaccharides. These oligosaccharides are known for their anti-inflammatory, antioxidant, and inhibitory effects on the proliferation of pathogenic cells [38]. Chondroitin lyase, along with certain hyaluronic acids, is capable of cleaving chondroitin sulfate into its constituent chondroitin sulfate oligosaccharides. These oligosaccharides exert inhibitory actions on inflammation and demonstrate potent antioxidant properties [39]. Various biomedical and pharmaceutical applications have also shown the antimicrobial activity of hyaluronic acid, and most of the hyaluronidase-producing microorganisms are Gram-positive bacteria [40]. Moreover, the EfaA gene that may play an adhesin role in endocarditis, the GroEL gene involved in the adhesion or invasion of various target cells or tissues, and clpC, clpE, and clpP genes, which are related to stress survival, respond to stress and protect proteins from excessive damage, also were annotated in the whole genome of *L. rhamnosus* CP-1 [41,42,43,44,45,46,47,48]. The gene encoding the hyaluronic acid capsule, cpsA, uppS, and gndA genes encoding the capsule, as well as the white gene encoding LPS, were also annotated. The hyaluronic acid capsule not only prevents phagocytosis by discouraging C3b binding, but also camouflages the bacteria as ‘self’ to the immune system, contributing to host immune evasion [49,50,51]. Moreover, LPS, apart from its role in immunity, is also involved in inflammatory signalling pathways [52,53]. These discoveries indicate that *L. rhamnosus* CP-1 may have anti-inflammatory potential at the molecular level.

The antimicrobial peptide ABC transporter permease component has been meticulously annotated within the KEGG database, while the ABC-type antimicrobial peptide transporter, which serves as the primary active component in the transport mechanism, has been documented in the TCDB. AMPs are short peptides consisting of 10 to 50 amino acids with a wide range of antimicrobial activities. They serve as crucial immune effector molecules, playing pivotal roles in initiating and modulating the host’s immune defence system to combat pathogenic bacteria [54]. The GlnP and GlnQ genes that contribute to bacterial survival in an acidic environment, as well as the clpL, and oppABCDF genes that contribute to acid and bile tolerance and bile salt resistance, were annotated in this *L. rhamnosus* CP-1 genome [55,56]. Activation of dltA gene expression protects LTA-expressing Gram-positive bacteria from the innate immune antimicrobial defence. The dltB and dltD genes may play significant roles in both human immune responses and anti-inflammatory processes [57]. Cellular metabolism can lead to excessive production of reactive oxygen species (ROS) and reactive nitrogen species (RNS), which induce oxidative stress and subsequently destroy macromolecules, and the clpC, clpB, and clpX genes respond to stress and protect proteins from excessive damage [58]. Among the metabolic pathways annotated in the KEGG database, terpenoid skeletons are involved in the biosynthesis of 10 genes, while keto-glycan units contribute to the biosynthesis of 5 genes. Medicinal plants, which are rich in compounds such as terpenoids, flavonoids, and phenols, have been shown to be effective in alleviating acute lung injury (ALI) or chronic obstructive pulmonary disease (COPD). Certain diterpenoids and triterpenoids demonstrated inhibitory activity against inflammatory diseases, with mechanistic studies indicating that their modulatory effects primarily occur through dual inhibition of the mitogen-activated protein kinase (MAPK) cascade and nuclear factor kappa-B (NF-κB) transcriptional activation pathways. Flavonoids, a class of polyphenolic compounds, exhibit biological activities including antihepatotoxic, anti-inflammatory, and anti-ulcer properties. Furthermore, numerous flavonoid derivatives have been demonstrated to possess therapeutic efficacy against pulmonary inflammation. As well-established anti-inflammatory phytochemicals, specific flavonoids have shown suppressive effects in various animal models of inflammation. The underlying mechanism of this inhibition may involve the attenuation of oxidative stress, suppression of NF-κB activation, and inhibition of epidermal growth factor receptor (EGFR) phosphorylation [59,60]. Additionally, *L. rhamnosus* CP-1 was isolated from the stools of mice with high-DHA tuna oil-relieved smoking-induced pneumoni [1]. In summary, whole-genome sequencing revealed that the *L. rhamnosus* CP-1 strain isolated in this study exhibits inherent acid and bile tolerance at the molecular level and possesses potential antimicrobial and anti-inflammatory properties at the genetic level. The subsequent phase may involve conducting specific in vivo or in vitro anti-inflammatory experiments based on the characteristics demonstrated at the molecular level. For instance, animal treatment model experiments could be implemented to further validate the anti-inflammatory functions associated with this strain.

## 5. Conclusions

In this study, we employed WGS technology to analyse the genomic characteristics of *L. rhamnosus* CP-1, a strain isolated from the gut of mice with pneumonia alleviated by probiotics. The results indicate that the genome of *L. rhamnosus* CP-1 consists of only one circular chromosome and no plasmid, the total length of *L. rhamnosus* CP-1 is 2,989,570 bp, and the GC content is 46.76%. It contains metabolic regulatory proteins and bacteriocin-related genes. Genomic subsystems analysis showed that *L. rhamnosus* CP-1 contained a non-toxic virulence factor, with 27 resistance genes and 112 carbohydrate-active enzyme genes annotated. The exploration of genome function revealed a wealth of annotations across multiple databases, with the GO database assigning labels to 1905 genes for biological processes, 1841 genes for molecular functions, and 588 genes for cellular components. Similarly, the COG database indexed 2334 genes, while the KEGG database encompassed 2700 genes. The Swiss-Prot database provided annotations for 1927 genes, and the TCDB listed 568 genes. Whole-genome sequencing analysis revealed that *L. rhamnosus* CP-1 possesses a robust genetic repertoire for carbohydrate metabolism and demonstrates potential anti-inflammatory properties at the molecular level. Potential functional genes include carbohydrate transport and hydrolase, antimicrobial peptide ABC transporter and its osmotic enzyme components, bacteriocin immune protein, terpenoid skeleton, and keto-glycan synthesis. Therefore, this study presents the whole-genome sequence analysis of *L. rhamnosus* CP-1 in detail, elucidating its potential for biotechnological applications. It thereby lays a solid foundation for further exploration of this bacterium’s probiotic effects and provides a theoretical basis for future research and development. However, the specific antibacterial and anti-inflammatory effects require further investigation through relevant experimental studies, both in vivo and in vitro.

## Figures and Tables

**Figure 1 biotech-14-00047-f001:**
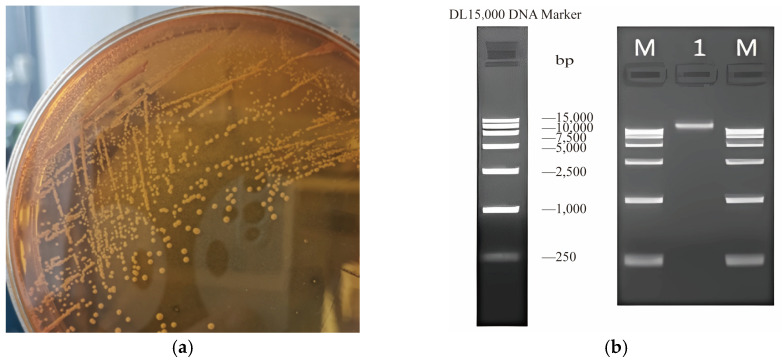
Identification of *L. rhamnosus* CP-1. (**a**) Orphological characteristics of *L. rhamnosus* CP-1; (**b**) results of agarose gel electrophoresis assay. Note: M is marker band and 1 is L. plantarum CP-1 band.

**Figure 2 biotech-14-00047-f002:**
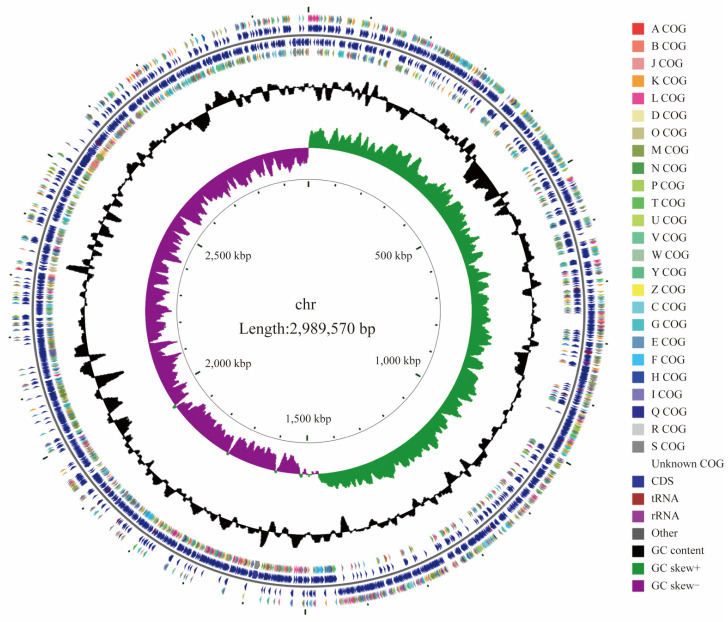
Genomic graphical map.

**Figure 3 biotech-14-00047-f003:**
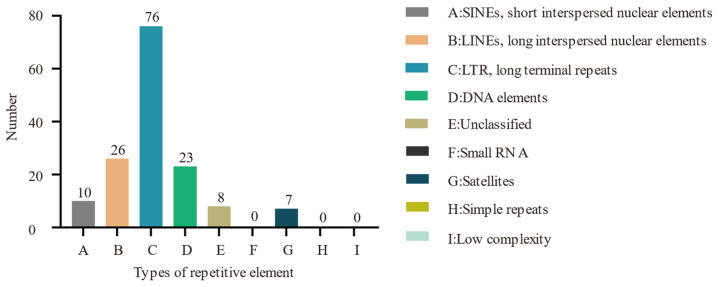
Distribution of repeated sequence elements.

**Figure 4 biotech-14-00047-f004:**
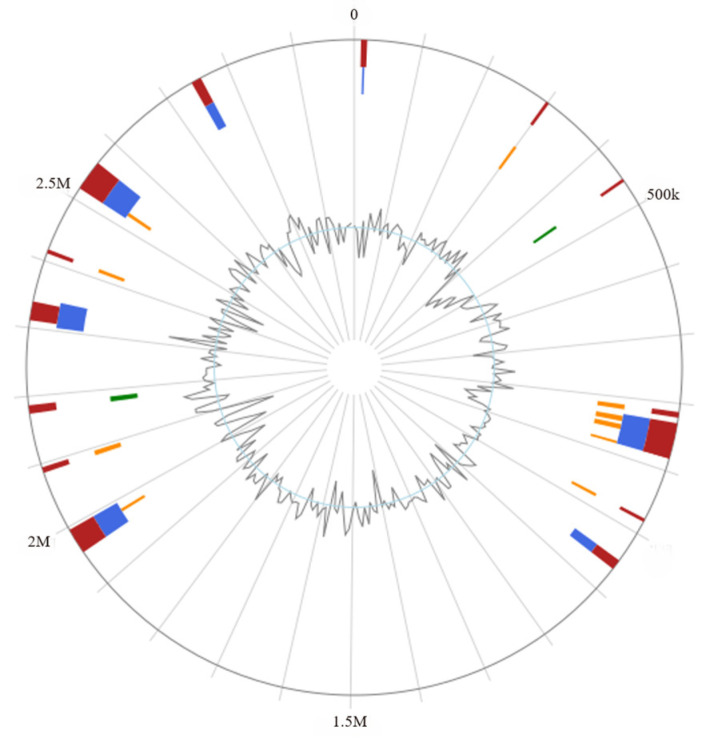
Genetic island prediction circle chart. Note: Red represents Predicted by at least one method; blue represents Predicted by IslandPath-DIMOB; yellow represents Predicted by SIGI-HMM; green represents Predicted by IslandPick; and red represents Predicted by at least one method Light blue represents the prediction of the Islander.

**Figure 5 biotech-14-00047-f005:**
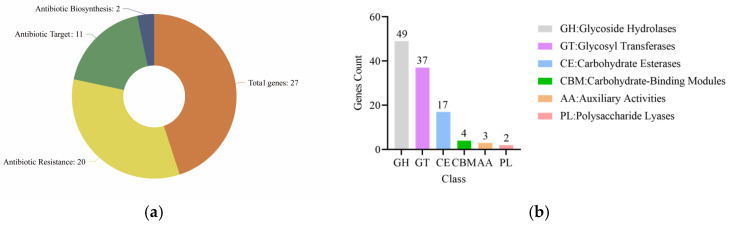
Genome subsystem analysis. (**a**) Distribution of antibiotic resistance genes; (**b**) distribution of carbohydrate-active enzymes.

**Figure 6 biotech-14-00047-f006:**
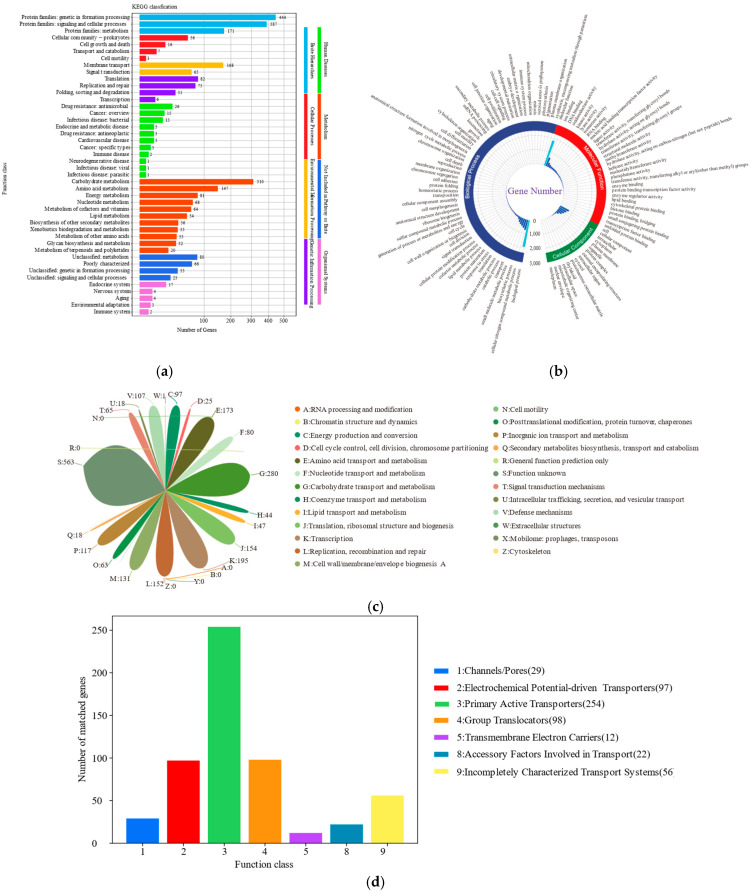
Genome functional annotation. (**a**) Genome KEGG functional annotation; (**b**) functional annotation of genomic GO. (**c**) Genome COG functional annotation. (**d**) TCDB database annotation: transporter protein distribution.

**Figure 7 biotech-14-00047-f007:**
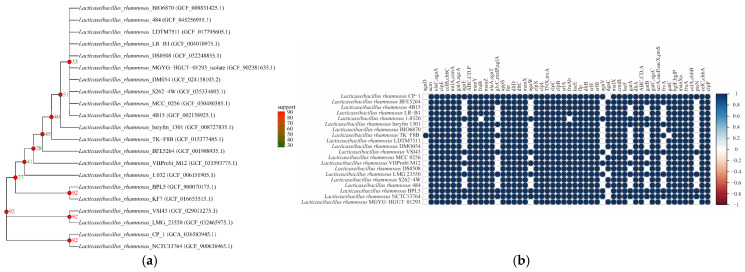
(**a**) Evolutionary tree of core gene. (**b**) Cluster analysis of functional genes.

**Table 1 biotech-14-00047-t001:** Overview of non-coding genes of *L. rhamnosus* CP-1.

Type	Copy Number	Avg. Length (bp)	Total Length (bp)	Percent of Genome (%)
5S rRNA	5	112	560	0.02
16S rRNA	5	1566	7830	0.26
23S rRNA	5	2914	14,570	0.49
tRNA	59	75	4467	0.15
ncRNA	39	176	6878	0.23

**Table 2 biotech-14-00047-t002:** Summary of prediction results of *L. rhamnosus* CP-1 prophages.

Prophage ID	Start	Stop	Start of attL	End of attL	Start of attR	End of attR
pp1	230,969	234,634	230,300	230,312	231,608	231,620
pp2	372,779	405,684	371,641	371,652	403,472	403,483
pp3	826,529	874,666	826,373	826,386	872,384	872,397
pp4	949,650	970,153	952,071	952,086	966,560	966,575
pp5	1,041,202	1,060,747	1,043,623	1,043,638	1,058,112	1,058,127
pp6	1,456,399	1,465,511	1,457,693	1,457,750	1,475,738	1,475,795
pp7	1,617,514	1,688,938	1,618,278	1,618,291	1,687,483	1,687,496
pp8	1,724,078	1,764,150	1,723,997	1,724,011	1,764,017	1,764,031
pp9	2,508,453	2,533,973	2,511,415	2,511,461	2,541,070	2,541,116

**Table 3 biotech-14-00047-t003:** Distribution of virulence factors.

VFDB ID	ORF Name	VFDB Name
VFG006717 (gb|NP_465159)	chr_718	VF0444
VFG000077 (gb|NP_465991)	chr_929	VF0074
VFG000964 (gb|WP_010922799)	chr_1067	VF0244
VFG046465 (gb|WP_003028672)	chr_1286	VF0460
VFG002190 (gb|WP_002362225)	chr_1528	VF0361
VFG048830 (gb|WP_014907233)	chr_1626	VF0560
VFG000080 (gb|NP_464522)	chr_1714	VF0073
VFG047039 (gb|WP_003018140)	chr_1939	VF0542
VFG012095 (gb|WP_003435012)	chr_2115	VF0594
VFG002165 (gb|WP_002356954)	chr_2293	VF0354
VFG000079 (gb|NP_463763)	chr_2370	VF0072

**Table 4 biotech-14-00047-t004:** Swiss-Prot database annotation results.

Hit_Name	Hit_Description	Hit_Length
Q9RGG2	PTS system sorbose-specific EIID component OS, Cell membrane, Multi-pass membrane protein	282
Q03AL7	UvrABC system protein B OS, Cytoplasm	671
B3WE47	UvrABC system protein C OS, Cytoplasm	602
P35855	Teichoic acid D-alanyltransferase OS, Cell membrane, Multi-pass membrane protein	405
Q03CT6	Adenylosuccinate synthetase OS, Cytoplasm	431
Q9RGG3	PTS system sorbose-specific EIIC component OS, Cell membrane, Multi-pass membrane protein	277
Q039N7	Peptide deformylase OS	184
Q033U7	Argininosuccinate synthase OS, Cytoplasm	406
Q9RGG4	PTS system sorbose-specific EIIB component OS, Cytoplasm	164
Q037U3	Thiamine transporter ThiT OS, Cell membrane, Multi-pass membrane protein	201
Q039K4	Trigger factor OS	445
Q9RGG5	PTS system sorbose-specific EIIA component OS, Cytoplasm	138
B3WAR0	Alanine racemase OS	378
B3WEY4	Bifunctional protein FolD OS	283
B3WEW9	Methionyl-tRNA formyltransferase OS	318
F9US27	Gallate decarboxylase OS	490
P11502	PTS system lactose-specific EIIA component OS, Cytoplasm	112
Q034S0	Large-conductance mechanosensitive channel OS, Cell membrane, Multi-pass membrane protein	123
Q38YD9	Protein RecA OS, Cytoplasm	355
Q88YI7	UvrABC system protein A OS, Cytoplasm	951
P71478	Cold shock protein 1 OS, Cytoplasm	66
P0A355	Cold shock-like protein CspLA OS, Cytoplasm	66
Q38XX6	Glycerol kinase OS	505
F9UTW9	Glycerol uptake facilitator protein 3 OS, Cell membrane, Multi-pass membrane protein	240
P71479	Bifunctional protein PyrR 1 OS	180

## Data Availability

The data sets utilised in the present study are archived in online repositories. The specific repository and its associated accession number(s) are detailed under PRJNA1069864.

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
