# Peer review of "Anti-Inflammatory Function Analysis of Lacticaseibacillus rhamnosus CP-1 Strain Based on Whole-Genome Sequencing"

_biotech, 2025, doi:10.3390/biotech14020047_

Round 1
Reviewer 1 Report
Comments and Suggestions for Authors
The authors report some bioinformatics results from the genome sequencing of an isolate of L. rhamnosus coming from mice used in a previous study. The paper reports on bioinformatics analyses performed to characterise the genome sequence, but the reporting should be improved:
- Over-statement: The title and conclusions make strong claims about anti-inflammatory properties that aren't supported by the data. The claims must be reduced, as there is no evidence that CP-1 actually reduces inflammatory markers. The paper requires major rewriting.
- Comparative genomics: this is one of the many Lacticaseibacillus rhamnosus genomes isolated and sequenced. In a paper with a strong focus on bioinformatics, I would expect phylogenetic analysis and core genome analysis with multiple strains. This must be reviewed. This could help put the isolate in perspective.
- Data availability: the bioproject only contains the assembled genomes, but I couldnt find the raw FASTQ files. Must be addressed.
Author Response
Reviewer #1: (Highlighted in Red)
The authors report some bioinformatics results from the genome sequencing of an isolate of L. rhamnosus coming from mice used in a previous study. The paper reports on bioinformatics analyses performed to characterise the genome sequence, but the reporting should be improved:
- Over-statement: The title and conclusions make strong claims about anti-inflammatory properties that aren't supported by the data. The claims must be reduced, as there is no evidence that CP-1 actually reduces inflammatory markers. The paper requires major rewriting.
Reply: Thank you for your comments. In accordance with your suggestions, the necessary corrections to the previous manuscript have been made, and changes and additions have been incorporated into the revised manuscript. The detailed corrections are listed below. Firstly, the article title has been revised from “Whole-Genome Sequencing Analysis Reveals Anti-inflammatory Properties of Lacticaseibacillus rhamnosus CP-1” to “Anti-inflammatory Functional Analysis of Lacticaseibacillus rhamnosus CP-1 Strain Based on Whole-Genome Sequencing” (Lines 2-3).
Then, the conclusion in the article abstract has been corrected into “Conclusions: Whole-genome sequencing analysis revealed that L. rhamnosus CP-1 has carbohydrate utilization and potential anti-inflammatory effects at the molecular level. Potential functional genes include carbohydrate transport and hydrolase, antimicrobial peptide ABC transporter and its osmotic enzyme components, bacteriocin immune protein, terpenoid skeleton, and keto-glycan synthesis” (Lines 25-29).
Finally, some of the concluding statements of the article and the conclusion at the end of the article have been made necessary modifications. Specific modifications:
1) These findings suggest that L.rhamnosus CP-1 has possible potential antibacterial activity at the molecular level, but the specific antibacterial properties need to be further verified by in vitro related antibacterial experiments (Lines 460-462).
2) These discoveries indicate that L.rhamnosus CP-1 may have anti-inflammatory potential at the molecular level (Lines 480-481).
3) In summary, whole-genome sequencing revealed that the L. rhamnosus CP-1 strain isolated in this study exhibits inherent acid and bile tolerance at the molecular level and possesses potential antimicrobial and anti-inflammatory properties at the genetic level (Lines 514-517).
5) Whole-genome sequencing analysis revealed that L. rhamnosus CP-1 possesses a robust genetic repertoire for carbohydrate metabolism and demonstrates potential anti-inflammatory properties at the molecular level. Potential functional genes include carbohydrate transport and hydrolase, antimicrobial peptide ABC transporter and its osmotic enzyme components, bacteriocin immune protein, terpenoid skeleton, and keto-glycan synthesis (Lines 535-540).
- Comparative genomics: this is one of the many Lacticaseibacillus rhamnosus genomes isolated and sequenced. In a paper with a strong focus on bioinformatics, I would expect phylogenetic analysis and core genome analysis with multiple strains. This must be reviewed. This could help put the isolate in perspective.
Reply: Thank you for your comments. At your suggestion, we have added sections on comparative genomics to the methods, results, and discussion sections, as follows:
1) 2.4 Comparative Genomic Analysis
Based on the chromosomal sequences derived from the genome assembly, we em-ployed the fastANI software to identify 20 phylogenetically proximate species within the database, subsequently performing phylogenetic reconstruction of sample core genes through the UBCG software. Functional genes were retrieved from the KEGG database. Then, L. rhamnosus CP - 1 was compared and analyzed with 20 other strains. The Pearson correlation coefficient was used to calculate the genetic correlations among different strains. A clustering algorithm was adopted to cluster the strains, and GraphPad Prism was employed to draw the graph (Lines 152-160).
2) 3.6 Comparative genomics of Lactobacillus rhamnosus CP-1 strain
According to the results of phylogenetic tree analysis, the phylogenetic relationship between L. rhamnosus CP-1 and other strains showed obvious hierarchical characteristics, which provided important clues for analyzing the evolutionary history of this strain. Lacticaseibacillus rhamnosus NCTC13764 and L. rhamnosus CP-1 clustered in the same branch, and the support value reached 92. The higher support value indicated that there was a close genetic relationship between them at the core gene level, this close clustering relationship probably reflects that they have a closer common ancestor or have experienced fewer genetic divergence events during evolution. Strains such as Lacticaseibacillus rhamnosus VSI43 and Lacticaseibacillus rhamnosus LMG 23550 were in a branch with higher support values with L. rhamnosus CP-1, showing a relatively close evolutionary relationship, based on this, it can be inferred that these strains may belong to the same evolutionary branch or subspecies, and they may share similar genetic background and evolutionary path during the long-term evolution, therefore, there is a high consistency in the composition and arrangement of core genes. However, other strains, such as Lacticaseibacillus rhamnosus BIO6870 and Lacticaseibacillus rhamnosus 484, were distributed in distant nodes and had significantly lower support values, which suggested that they were genetically different from L. rhamnosus CP-1, these differences may be due to different selection pressures during evolution, differences in horizontal gene transfer events, or long-term geographical isolation, which may lead to their different subpopulations (Fig. 7a). The results of functional gene cluster analysis showed that Most strains showed dark blue color with L. rhamnosus CP-1 at many gene loci, such as agaD, acmA, etc. A small number of white loci means that there is a weak correlation with L. rhamnosus CP-1 strain at the corresponding gene locus, and there are differences in genetic characteristics. Strains such as Lacticaseibacillus rhamnosus BFE5264 and 4B15 were similar to L. rhamnosus CP-1 in most genetic loci, but showed white in some loci, showing some genetic differences (Fig. 7b) (Lines 341-367).
(a) |
(b) |
Figure 7. (a) Evolutionary Tree of core gene. (b) Cluster analysis of functional genes.
3) In addition, according to the results of genome alignment, it was indicated that the unique evolutionary status of L. rhamnosus CP-1 was revealed, which formed a close association cluster with a specific closely related strain on the core gene evolutionary tree, indicating that L. rhamnosus CP-1 is an important genetic resource for L. rhamnosus, it not only reflects the high genetic similarity with some strains, but also clarifies the genetic boundaries with other strains, this study provided a solid theoretical basis for further study on the phylogeny, population differentiation and evolution of functional genes of the strain. At the same time, functional gene cluster analysis showed that strain zero exhibited strong correlation and consistent genetic characteristics with L. rhamnosus CP-1 at many gene loci (such as agaD, ACMA, etc.). A small number of genetic loci showed weak correlation, indicating that there were differences in genetic characteristics. Most of these genes are related to the synthesis of sugar-specific components of glucose, sorbose, cellobiose, fructose, and lactose, which can produce SCFAs through related metabolic pathways, and they are also involved in the regulation of glucose metabolism, and play a variety of anti-inflammatory and antibacterial-related physiological functions. This genetic difference may affect the functional characteristics of strains, which provides important clues for further study of genetic diversity and functional differences between strains (Lines 394-411).
- Data availability: the bioproject only contains the assembled genomes, but I couldnt find the raw FASTQ files. Must be addressed.
Reply: Thank you for your comments. In accordance with your recommendation, we have re-uploaded the original FASTQ files to the repository while maintaining the original accession number PRJNA1069864.

Reviewer 2 Report
Comments and Suggestions for Authors
This study reports the whole-genome sequencing of Lacticaseibacillus rhamnosus CP-1, a probiotic strain isolated from fish oil-treated, smoking-induced pneumonic mice. Genomic analysis revealed genes linked to carbohydrate metabolism, antimicrobial activity, stress tolerance, and potential anti-inflammatory functions. The findings suggest CP-1 may be a promising strain for probiotic applications, pending further functional validation.
Questions from my end are:
Have you validated the predicted anti-inflammatory genes functionally?While the genome annotation suggests anti-inflammatory potential (e.g., dltA, clpC, hyaluronidase), there’s no experimental data (gene expression, cytokine assays, etc.) to prove these genes are active or impactful under stress or inflammation.
Did you compare CP-1 to reference probiotic strains like L. rhamnosus GG? Without a genomic comparison to established strains, it's hard to tell what's unique or novel about CP-1. A table or phylogenetic analysis would help highlight its advantages.
What is the phenotypic profile for antibiotic resistance genes (ARGs)? You mention 27 ARGs, but no phenotypic tests (like MICs) are shown. This is critical for confirming safety—especially if CP-1 is intended for food or therapeutic use.
Are any ARGs or virulence genes located in prophages or genomic islands? Prophages and gene islands can mediate horizontal gene transfer. It’s essential to rule out the risk of spreading resistance or virulence traits to other microbes.
What are the key genome assembly quality metrics (N50, coverage, completeness), These metrics confirm whether the genome is accurately assembled and complete. Without them, it's unclear if downstream annotations are reliable.
Have you tested the strain’s ability to metabolize specific carbohydrates? Genomic data shows extensive carbohydrate metabolism (PTS system, CAZy genes), but no actual sugar utilization assays were done. This is needed to link genotype to functional probiotic use.
What safety evaluations have been done beyond WGS? You report no plasmids and low virulence genes, which is good. But in vitro assays for bile/acid tolerance, immune modulation, or cytotoxicity would give stronger safety assurance.
Can CP-1 inhibit pathogens? Has bacteriocin activity been tested? Bacteriocin-related genes were identified, but no inhibition assay (e.g., against E. coli, Salmonella) was shown to confirm functional antibacterial potential.
Are the predicted terpenoid and glycan biosynthesis pathways active? These genes are rare and exciting in L. rhamnosus. If active, they could lead to new bioactive metabolites. A basic metabolomics screen would help validate.
What distinguishes CP-1 as a probiotic over other native gut strains? Since CP-1 was isolated from fish-oil-fed mice, it’s unclear if it’s a beneficial result of treatment or a pre-existing commensal. Control strain comparisons would help clarify this.
Author Response
Reviewer #2: (Highlighted in Green)
This study reports the whole-genome sequencing of Lacticaseibacillus rhamnosus CP-1, a probiotic strain isolated from fish oil-treated, smoking-induced pneumonic mice. Genomic analysis revealed genes linked to carbohydrate metabolism, antimicrobial activity, stress tolerance, and potential anti-inflammatory functions. The findings suggest CP-1 may be a promising strain for probiotic applications, pending further functional validation.
Questions from my end are:
- Have you validated the predicted anti-inflammatory genes functionally?While the genome annotation suggests anti-inflammatory potential (e.g., dltA, clpC, hyaluronidase), there’s no experimental data (gene expression, cytokine assays, etc.) to prove these genes are active or impactful under stress or inflammation.
Reply: Thank you for your comments. This study obtained relevant anti-inflammatory genes through whole genome prediction analysis technology screening. Currently, the team is conducting inflammation and anti-inflammatory effect verification experiments based on animal models. The systematic paper writing work of the related research results has been launched at the same time. The anti-inflammatory mechanism analysis and potential application value of this part of the gene function will be comprehensively elaborated in subsequent study.
- Did you compare CP-1 to reference probiotic strains like L. rhamnosus GG? Without a genomic comparison to established strains, it's hard to tell what's unique or novel about CP-1. A table or phylogenetic analysis would help highlight its advantages.
Reply: Thank you for your comments. Following your suggestion, we have added a comparative analysis of the core genome of this strain to the methods, results, and discussion sections, as follows:
1) 2.4 Comparative Genomic Analysis
Based on the chromosomal sequences derived from the genome assembly, we em-ployed the fastANI software to identify 20 phylogenetically proximate species within the database, subsequently performing phylogenetic reconstruction of sample core genes through the UBCG software (Lines 152-156).
2) 3.6 Comparative genomics of Lactobacillus rhamnosus CP-1 strain
According to the results of phylogenetic tree analysis, the phylogenetic relationship between L. rhamnosus CP-1 and other strains showed obvious hierarchical characteristics, which provided important clues for analyzing the evolutionary history of this strain. Lacticaseibacillus rhamnosus NCTC13764 and L. rhamnosus CP-1 clustered in the same branch, and the support value reached 92. The higher support value indicated that there was a close genetic relationship between them at the core gene level, this close clustering relationship probably reflects that they have a closer common ancestor or have experienced fewer genetic divergence events during evolution. Strains such as Lacticaseibacillus rhamnosus VSI43 and Lacticaseibacillus rhamnosus LMG 23550 were in a branch with higher support values with L. rhamnosus CP-1, showing a relatively close evolutionary relationship, based on this, it can be inferred that these strains may belong to the same evolutionary branch or subspecies, and they may share similar genetic background and evolutionary path during the long-term evolution, therefore, there is a high consistency in the composition and arrangement of core genes. However, other strains, such as Lacticaseibacillus rhamnosus BIO6870 and Lacticaseibacillus rhamnosus 484, were distributed in distant nodes and had significantly lower support values, which suggested that they were genetically different from L. rhamnosus CP-1, these differences may be due to different selection pressures during evolution, differences in horizontal gene transfer events, or long-term geographical isolation, which may lead to their different subpopulations (Fig. 7a) (Lines 341-360).
(a) |
Figure 7. (a) Evolutionary Tree of core gene.
3) In addition, according to the results of genome alignment, it was indicated that the unique evolutionary status of L. rhamnosus CP-1 was revealed, which formed a close association cluster with a specific closely related strain on the core gene evolutionary tree, indicating that L. rhamnosus CP-1 is an important genetic resource for L. rhamnosus, it not only reflects the high genetic similarity with some strains, but also clarifies the genetic boundaries with other strains, this study provided a solid theoretical basis for further study on the phylogeny, population differentiation and evolution of functional genes of the strain (Lines 394-401).
- What is the phenotypic profile for antibiotic resistance genes (ARGs)? You mention 27 ARGs, but no phenotypic tests (like MICs) are shown. This is critical for confirming safety—especially if CP-1 is intended for food or therapeutic use.
Reply: Thank you for your comments. As for the follow-up verification of the antibiotic resistance gene phenotype experiment you mentioned, we have carried out corresponding experiments, and are currently conducting systematic collation and in-depth analysis of relevant experimental data, the writing of the new article is expected to be completed soon. Selected antibiotics included streptomycin, erythromycin, kanamycins, chloramphenicol and Ampicillin. The minimum inhibitory concentration (MIC) of each antibiotic was determined by twofold serial dilution procedures in broth, and the antibiotic resistance of the strain was evaluated by comparing the resulting MIC values with the published values. Some of the results are as follows:
antibiotics |
MIC(μg/mL) |
streptomycin |
32 |
erythromycin |
0.5 |
chloramphenicol |
1 |
ampicillin |
2 |
However, these data will be presented in subsequent studies. Since this study focuses on predicting the activity of the strain based on whole-genome sequencing, the relevant results are not included in this study.
- Are any ARGs or virulence genes located in prophages or genomic islands? Prophages and gene islands can mediate horizontal gene transfer. It’s essential to rule out the risk of spreading resistance or virulence traits to other microbes.
Reply: Thank you for your comments. No ARGs or virulence genes were found in prophages. One virulence gene associated with bacteriocin and four antibiotic resistance genes (three linked to bacteriocin and one related to ABC transporter) were identified within genomic islands. Therefore, there is no risk of transmission of undesirable traits. The above results have been added to section 3.4 in the revised manuscript (Lines 258-260).
- What are the key genome assembly quality metrics (N50, coverage, completeness), These metrics confirm whether the genome is accurately assembled and complete. Without them, it's unclear if downstream annotations are reliable.
Reply: Thank you for your comments. For the sequencing and assembly of the genome, we used 5 maps, including single base mass map, base content map, GC content map, sequence base quality profiles and Sequence length profiles of third-generation sequencing data (Supplementary Fig. S1). In addition, as suggested by you, we have added a sequence length table for the third-generation sequencing data. These results indicate that the overall quality of the data is very high and can be used for subsequent compositional and assembly outcome-based predictions. The sequence length statistics for the third-generation sequencing data are as follows (Line 575).
Sample |
Seq Type |
Total sequence number |
219,663 |
Total sequence length (bp) |
1,957,878,605 |
Longest (bp) |
144,049 |
Shortest (bp) |
1 |
N20 (bp) |
27,196 |
N50 (bp) |
16,004 |
N90 (bp) |
4,519 |
N number |
0 |
N rate |
0 |
GC content % |
46.51 |
- Have you tested the strain’s ability to metabolize specific carbohydrates? Genomic data shows extensive carbohydrate metabolism (PTS system, CAZy genes), but no actual sugar utilization assays were done. This is needed to link genotype to functional probiotic use.
Reply: Thank you for your comments. This study aims to investigate the potential biological functions and characteristics of L. rhamnosus CP-1 at the molecular level through whole-genome sequencing technology, with particular focus on exploring its potential anti-inflammatory mechanisms at the genetic level. We have conducted corresponding animal experiments on the actual sugar utilization determination of this strain, and are currently systematically organizing and analyzing the relevant experimental data. We expect to complete the writing of a new article soon.
- What safety evaluations have been done beyond WGS? You report no plasmids and low virulence genes, which is good. But in vitro assays for bile/acid tolerance, immune modulation, or cytotoxicity would give stronger safety assurance.
Reply: Thank you for your comments. Based on your suggestion, we plan to implement rigorous in vitro evaluations of bile/acid tolerance, immunomodulatory properties, and cytotoxic effects through standardized bioassays to systematically characterize the strain's biosafety profile. Concurrently, our research team is currently conducting inflammation and anti-inflammatory effect verification experiments based on animal models, aiming to deeply evaluate its anti-inflammatory activity and mechanism of action through pathological indicators and immunohistochemical analysis. At the same time, we have started systematic organization and data verification of existing research results, and the writing of academic papers has also been synchronized. The relevant content will be comprehensively elaborated in subsequent research papers.
- Can CP-1 inhibit pathogens? Has bacteriocin activity been tested? Bacteriocin-related genes were identified, but no inhibition assay (e.g., against E. coli, Salmonella) was shown to confirm functional antibacterial potential.
Reply: Thank you for your comments. Regarding the research progress of the antibacterial experiment of the strain, the research team has preliminarily completed the implementation of the main experimental part and is currently conducting systematic data sorting and analysis. It should be noted that due to its unique experimental orientation, the research data in this part will be integrated into other relevant experimental research papers for in-depth interpretation. This arrangement will not affect the independence of the experimental data and the completeness of the conclusions in the current submitted paper. Some of the results of the inhibition of the strain against pathogenic bacteria are shown in the diagram below:
Note, Vs, Vibrio splendidus; Pp, Pseudomonasputida; Ah, Aeromonas hydrophild;
Vh,Vibrio harveyi; Ab, Acinetobacter baumannii; Vm, Vibrio mimicus.
- Are the predicted terpenoid and glycan biosynthesis pathways active? These genes are rare and exciting in L. rhamnosus. If active, they could lead to new bioactive metabolites. A basic metabolomics screen would help validate.
Reply: Thank you for your comments. In the KEGG pathway database, 20 genes associated with polyketide and sugar unit biosynthesis as well as terpenoid backbone biosynthesis have been annotated within the metabolic category. And we will follow up with recommendations for metabolomic screening.
- What distinguishes CP-1 as a probiotic over other native gut strains? Since CP-1 was isolated from fish-oil-fed mice, it’s unclear if it’s a beneficial result of treatment or a pre-existing commensal. Control strain comparisons would help clarify this.
Reply: Thank you for your comments. According to your suggestion, we added the comparative analysis of related functional genes of this strain in the methods, results and discussion section. The specific results are as follows:
Functional genes were retrieved from the KEGG database. Then, L. rhamnosus CP -1 was compared and analyzed with 20 other strains. The Pearson correlation coefficient was used to calculate the genetic correlations among different strains. A clustering al-gorithm was adopted to cluster the strains, and GraphPad Prism was employed to draw the graph (Lines 156-160).
The results of functional gene cluster analysis showed that Most strains showed dark blue color with L. rhamnosus CP-1 at many gene loci, such as agaD, acmA, etc. A small number of white loci means that there is a weak correlation with L. rhamnosus CP-1 strain at the corresponding gene locus, and there are differences in genetic characteristics. Strains such as Lacticaseibacillus rhamnosus BFE5264 and 4B15 were similar to L. rhamnosus CP-1 in most genetic loci, but showed white in some loci, showing some genetic differences (Fig. 7b) (Lines 360-367).
(b) |
Figure 7. (b) Cluster analysis of functional genes.
3) At the same time, functional gene cluster analysis showed that strain zero exhibited strong correlation and consistent genetic characteristics with L. rhamnosus CP-1 at many gene loci (such as agaD, ACMA, etc.). A small number of genetic loci showed weak correlation, indicating that there were differences in genetic characteristics. Most of these genes are related to the synthesis of sugar-specific components of glucose, sorbose, cellobiose, fructose, and lactose, which can produce SCFAs through related metabolic pathways, and they are also involved in the regulation of glucose metabolism, and play a variety of anti-inflammatory and antibacterial-related physiological functions. This genetic difference may affect the functional characteristics of strains, which provides important clues for further study of genetic diversity and functional differences between strains (Lines 401-411).

Reviewer 3 Report
Comments and Suggestions for Authors
Manuscript “Whole-Genome Sequencing Analysis Reveals Anti-inflammatory Properties of Lacticaseibacillus rhamnosus CP-1”
Comments:
- The authors mention isolating the strain from fish oil-treated mice but do not provide a rationale for choosing this specific strain over others. please discuss, explain why L. rhamnosus CP-1 was selected and how it differs functionally or genetically from other strains.
- While genomic data is extensive, no experimental validation (e.g., in vitro or in vivo anti-inflammatory assays) is presented in the results and discussion section. The author should have included or proposed at least preliminary biological testing to support predicted anti-inflammatory functions.
- KEGG, GO, and COG data are reported in large volumes without direct linkage to specific biological relevance or phenotypes. But authors have not focused key pathways related to inflammation or probiotic functionality and minimize redundant annotation statistics. Add this information.
- Add a comparative table or figure to show differences/similarities in genome structure or probiotic genes for a better understanding of the new findings in the study.
- The manuscript mentions terpenoid and keto-glycan biosynthesis genes but does not connect them clearly to probiotic or anti-inflammatory functions. The author needs to discuss how these pathways contribute to host interaction or immune modulation.
- Many figures and tables are not clearly referenced or discussed in the text. Please add.
- The conclusion needs to be rewritten by adding key findings and future suggestions.
Author Response
Reviewer #3: (Highlighted in Blue)
Comments:
- The authors mention isolating the strain from fish oil-treated mice but do not provide a rationale for choosing this specific strain over others. please discuss, explain why L. rhamnosus CP-1 was selected and how it differs functionally or genetically from other strains.
Reply: Thank you for your comments. According to your nice suggestions, we have made necessary corrections to our previous draft and changes/additions to the revised manuscript, the detailed corrections are listed below.
Firstly, the introductory section has been comprehensively revised to provide a rigorous rationale for the selection of L. rhamnosus, with substantive modifications implemented as follows:
1) In our previous study, the anti-inflammatory and antioxidant functions of tuna oil (TO) in cigarette smoke (CS) exposure-induced lung inflammation in mice were demonstrated. At the genus level, the abundance of Lacticaseibacillus in the CS-induced model group was reduced compared with the blank control group. The abundance of Lacticaseibacillus increased in the experimental group with different doses of fish oil when compared with the CS-induced pneumonic mice. At the species level, the abun-dance of Lacticaseibacillus rhamnosus (L. rhamnosus) increased after treatment with different doses of TO[1]. The anti-inflammatory properties of Lacticaseibacillus have been shown to enhance the bi-osynthesis of serum IL-10, while concurrently inhibiting the production of TNF-α, IL-6, and Interleukin-12 (IL-12). This dual action contributes to the subsequent mitigation of inflammatory responses in animals[2, 3]. L. rhamnosus, a widely studied Lacticaseibacillus in both domestic and international research. And it is a facultative anaerobic, non-spore-forming Gram-positive bacteria with long or short rods. It is classified within the genus Lacticaseibacillus, under the rhamnose subspecies. It is acid-resistant, resistant to pancreatic juice and bile salts, resistant to a variety of antibiotics and other biological characteristics[4]. L.rhamnosus is a symbiotic microorganism in the intestinal system of human and animals. Its advantage lies in its high intestinal adhesion rate and strong colonization. And it is beneficial to improve the host’s systemic immune response, is often used to enhance the body’s immunity and disease prevention and treatment[5] (Lines 34-52).
2) A growing body of evidence demonstrates that L. rhamnosus and its metabolites can modulate immune cells such as M1 macrophages and T lymphocytes, effectively suppressing the production of pro-inflammatory cytokines including tumour necrosis factor-alpha (TNF-α), interleukin-1β (IL-1β), interleukin-6 (IL-6), and interleukin-2 (IL-2). Concurrently, these microbial components enhance the generation of anti-inflammatory cytokines like interleukin-10 (IL-10), thereby regulating inflammatory responses and maintaining immune homeostasis through this dual mechanism of cytokine balance modulation[8-10] (Lines 55-62).
Secondly, we added sections on comparative genomics to the methods, results, and discussion sections, respectively, as follows:
1) 2.4 Comparative Genomic Analysis
Based on the chromosomal sequences derived from the genome assembly, we em-ployed the fastANI software to identify 20 phylogenetically proximate species within the database, subsequently performing phylogenetic reconstruction of sample core genes through the UBCG software. Functional genes were retrieved from the KEGG database. Then, L. rhamnosus CP - 1 was compared and analyzed with 20 other strains. The Pearson correlation coefficient was used to calculate the genetic correlations among different strains. A clustering algorithm was adopted to cluster the strains, and GraphPad Prism was employed to draw the graph (Lines 152-160).
2) 3.5 Comparative genomics of Lactobacillus rhamnosus CP-1 strain
According to the results of phylogenetic tree analysis, the phylogenetic relationship between L. rhamnosus CP-1 and other strains showed obvious hierarchical characteristics, which provided important clues for analyzing the evolutionary history of this strain. Lacticaseibacillus rhamnosus NCTC13764 and L. rhamnosus CP-1 clustered in the same branch, and the support value reached 92. The higher support value indicated that there was a close genetic relationship between them at the core gene level, this close clustering relationship probably reflects that they have a closer common ancestor or have experienced fewer genetic divergence events during evolution. Strains such as Lacticaseibacillus rhamnosus VSI43 and Lacticaseibacillus rhamnosus LMG 23550 were in a branch with higher support values with L. rhamnosus CP-1, showing a relatively close evolutionary relationship, based on this, it can be inferred that these strains may belong to the same evolutionary branch or subspecies, and they may share similar genetic background and evolutionary path during the long-term evolution, therefore, there is a high consistency in the composition and arrangement of core genes. However, other strains, such as Lacticaseibacillus rhamnosus BIO6870 and Lacticaseibacillus rhamnosus 484, were distributed in distant nodes and had significantly lower support values, which suggested that they were genetically different from L. rhamnosus CP-1, these differences may be due to different selection pressures during evolution, differences in horizontal gene transfer events, or long-term geographical isolation, which may lead to their different subpopulations (Fig. 7a). The results of functional gene cluster analysis showed that Most strains showed dark blue color with L. rhamnosus CP-1 at many gene loci, such as agaD, acmA, etc. A small number of white loci means that there is a weak correlation with L. rhamnosus CP-1 strain at the corresponding gene locus, and there are differences in genetic characteristics. Strains such as Lacticaseibacillus rhamnosus BFE5264 and 4B15 were similar to L. rhamnosus CP-1 in most genetic loci, but showed white in some loci, showing some genetic differences (Fig. 7b) (Lines 341-367).
(a) |
(b) |
Figure 7. (a) Evolutionary Tree of core gene. (b) Cluster analysis of functional genes.
3) In addition, according to the results of genome alignment, it was indicated that the unique evolutionary status of L. rhamnosus CP-1 was revealed, which formed a close association cluster with a specific closely related strain on the core gene evolutionary tree, indicating that L. rhamnosus CP-1 is an important genetic resource for L. rhamnosus, it not only reflects the high genetic similarity with some strains, but also clarifies the genetic boundaries with other strains, this study provided a solid theoretical basis for further study on the phylogeny, population differentiation and evolution of functional genes of the strain. At the same time, functional gene cluster analysis showed that strain zero exhibited strong correlation and consistent genetic characteristics with L. rhamnosus CP-1 at many gene loci (such as agaD, ACMA, etc.). A small number of genetic loci showed weak correlation, indicating that there were differences in genetic characteristics. Most of these genes are related to the synthesis of sugar-specific components of glucose, sorbose, cellobiose, fructose, and lactose, which can produce SCFAs through related metabolic pathways, and they are also involved in the regulation of glucose metabolism, and play a variety of anti-inflammatory and antibacterial-related physiological functions. This genetic difference may affect the functional characteristics of strains, which provides important clues for further study of genetic diversity and functional differences between strains (Lines 394-411).
- While genomic data is extensive, no experimental validation (e.g., in vitro or in vivo anti-inflammatory assays) is presented in the results and discussion section. The author should have included or proposed at least preliminary biological testing to support predicted anti-inflammatory functions.
Reply: Thank you for your comments. In response to the potential application value of this strain, our research team is currently conducting bidirectional validation experiments on its anti-inflammatory activity and inflammation regulation mechanism based on animal experimental models. Building upon these experimental findings, we have initiated the preparation of a comprehensive manuscript that will systematically integrate empirical data with theoretical frameworks. At that time, core scientific issues such as the molecular mechanism of action, dose-response relationship, and safety evaluation of the strain will be deeply explored and comprehensively explained. At the same time, based on your suggestions, we have supplemented in the revised draft the initial research ideas on the construction of biological experimental protocols for the systematic verification of the potential probiotic properties of the target strains, which will provide a reference for the future research of probiotics, it is expected to provide theoretical basis and methodological reference for subsequent in-depth research.
1) The subsequent phase may involve conducting specific in vivo or in vitro anti-inflammatory experiments based on the characteristics demonstrated at the molecular level. For instance, animal treatment model experiments could be implemented to further validate the anti-inflammatory functions associated with this strain (Line517-521).
2) Therefore, this study presents the whole genome sequence analysis of L. rhamnosus CP-1 in detail, elucidating its potential for biotechnological applications. It thereby lays a solid foundation for further exploration of this bacterium’s probiotic effects and provides a theoretical basis for future research and development. However, the specific antibacterial and anti-inflammatory effects require further investigation through relevant experimental studies, both in vivo and in vitro (Lines 540-545).
- KEGG, GO, and COG data are reported in large volumes without direct linkage to specific biological relevance or phenotypes. But authors have not focused key pathways related to inflammation or probiotic functionality and minimize redundant annotation statistics. Add this information.
Reply: Thank you for your comments. In accordance with your recommendations, we have endeavored to streamline the Discussion section by minimizing elaborations on annotation outcomes, while strategically redirecting analytical emphasis toward pivotal pathways associated with the strain's anti-inflammatory properties and probiotic potential. The substantive revisions have been implemented as follows:
The diversity of carbohydrate metabolism genes enables probiotics to efficiently degrade dietary fibers (e.g., cellulose, pectin, inulin) that are indigestible to the host, converting them into short-chain fatty acids (SCFAs, including acetate, propionate, and butyrate). These metabolites serve not only as crucial energy substrates for intestinal epithelial cells but also exert multiple physiological functions: lowering intestinal pH to inhibit pathogenic bacterial proliferation, activating host immune signaling pathways (e.g.,G protein-coupled receptors GPR41/43) to modulate local inflammatory responses and barrier functions, and systemically influencing host metabolism through circulatory transport[36]. Furthermore, the superior carbohydrate utilization capacity grants probiotics ecological competitiveness in the gut niche, allowing them to outcompete pathogens through nutrient sequestration, antimicrobial secretion, and adhesion-mediated colonization, thereby maintaining microbial homeostasis. Certain genes may additionally participate in metabolizing mucin-derived oligosaccharides within the intestinal mucus layer, facilitating mucus renewal and enhancing barrier integrity, while pathway byproducts such as B vitamins supplement host nutritional requirements. The synergistic effects of these functions not only enhance probiotic adaptability in the complex gut environment but also extend their health benefits through systemic metabolic regulation (e.g., SCFA-mediated modulation of hepatic and adipose tissue metabolism), potentially ameliorating metabolic disorders including obesity and diabetes[16, 34-36]. Notably, the enrichment of such genes may reflect co-evolutionary adaptations between probiotics and hosts during long-term symbiosis. However, comprehensive validation integrating transcriptomic/proteomic analyses and in vivo models remains essential to elucidate the functional activity and physiological contributions of these genetic elements (Lines 427-449).
- Add a comparative table or figure to show differences/similarities in genome structure or probiotic genes for a better understanding of the new findings in the study.
Reply: Thank you for your comments. According to your suggestion, we added the comparative analysis of related functional genes of this strain in the methods, results and discussion section. The specific results are as follows:
Functional genes were retrieved from the KEGG database. Then, L. rhamnosus CP -1 was compared and analyzed with 20 other strains. The Pearson correlation coefficient was used to calculate the genetic correlations among different strains. A clustering al-gorithm was adopted to cluster the strains, and GraphPad Prism was employed to draw the graph (Lines 156-160).
The results of functional gene cluster analysis showed that Most strains showed dark blue color with L. rhamnosus CP-1 at many gene loci, such as agaD, acmA, etc. A small number of white loci means that there is a weak correlation with L. rhamnosus CP-1 strain at the corresponding gene locus, and there are differences in genetic characteristics. Strains such as Lacticaseibacillus rhamnosus BFE5264 and 4B15 were similar to L. rhamnosus CP-1 in most genetic loci, but showed white in some loci, showing some genetic differences (Fig. 7b) (Lines 360-367).
(b) |
Figure 7. (b) Cluster analysis of functional genes.
3) At the same time, functional gene cluster analysis showed that strain zero exhibited strong correlation and consistent genetic characteristics with L. rhamnosus CP-1 at many gene loci (such as agaD, ACMA, etc.). A small number of genetic loci showed weak correlation, indicating that there were differences in genetic characteristics. Most of these genes are related to the synthesis of sugar-specific components of glucose, sorbose, cellobiose, fructose, and lactose, which can produce SCFAs through related metabolic pathways, and they are also involved in the regulation of glucose metabolism, and play a variety of anti-inflammatory and antibacterial-related physiological functions. This genetic difference may affect the functional characteristics of strains, which provides important clues for further study of genetic diversity and functional differences between strains (Lines 401-411).
- The manuscript mentions terpenoid and keto-glycan biosynthesis genes but does not connect them clearly to probiotic or anti-inflammatory functions. The author needs to discuss how these pathways contribute to host interaction or immune modulation.
Reply: Thank you for your comments. According to your nice suggestions, we focused in part of the discussion on establishing connections between terpenoid and ketose biosynthetic genes and the probiotic or anti-inflammatory properties of the strain, with particular emphasis on examining how these metabolic pathways may enhance host-microbe interactions or mediate immunomodulatory effects. Amend to read as follows:
Certain diterpenoids and triterpenoids have demonstrated inhibitory activity against inflammatory diseases. With mechanistic studies indicating their modulatory effects primarily occur through dual inhibition of the mitogen-activated protein kinase (MAPK) cascade and nuclear factor kappa-B (NF-κB) transcriptional activation pathways. Flavonoids, a class of polyphenolic compounds, exhibit biological activities including antihepatotoxic, anti-inflammatory, and anti-ulcer properties. Furthermore, numerous flavonoid derivatives have been demonstrated to possess therapeutic efficacy against pulmonary inflammation. As well-established anti-inflammatory phytochemicals, specific flavonoids have shown suppressive effects in various animal models of inflammation. The underlying mechanism of this inhibition may involve the attenuation of oxidative stress, suppression of NF-κB activation, and inhibition of epidermal growth factor receptor (EGFR) phosphorylation[59, 60] (Lines 502-513).
- Many figures and tables are not clearly referenced or discussed in the text. Please add.
Reply: Thank you for your comments. We feel sorry for our carelessness. Based on your comments, in our resubmitted manuscript, we have made the corresponding changes. Thanks for your correction.
- The conclusion needs to be rewritten by adding key findings and future suggestions.
Reply: Thank you for your comments. According to your nice suggestions, we have made the corresponding changes in the conclusion. The detailed revisions are as follows: Whole-genome sequencing analysis revealed that L. rhamnosus CP-1 possesses a robust genetic repertoire for carbohydrate metabolism and demonstrates potential anti-inflammatory properties at the molecular level. Potential functional genes include carbohydrate transport and hydrolase, antimicrobial peptide ABC transporter and its osmotic enzyme components, bacteriocin immune protein, terpenoid skeleton, and keto-glycan synthesis. Therefore, this study presents the whole genome sequence analysis of L. rhamnosus CP-1 in detail, elucidating its potential for biotechnological applications. It thereby lays a solid foundation for further exploration of this bacterium’s probiotic effects and provides a theoretical basis for future research and development. However, the specific antibacterial and anti-inflammatory effects require further investigation through relevant experimental studies, both in vivo and in vitro (Lines 535-545).

Round 2
Reviewer 2 Report
Comments and Suggestions for Authors
All the comments are addressed in detail and hence it can be accepted in the present form from my end.
Reviewer 3 Report
Comments and Suggestions for Authors
Authors have revised the manuscript nicely, hence, it can now be accepted for publication.